# Duration of inter-pregnancy interval and its predictors among pregnant women in urban South Ethiopia: Cox gamma shared frailty modeling

Belayneh Hamdela Jena[1,2]*, Gashaw Andargie Biks[3], Yigzaw Kebede Gete[2], Kassahun Alemu Gelaye[2]

1 Department of Public Health, College of Medicine and Health Sciences, Wachemo University, Hossana, Ethiopia, 2 Department of Epidemiology and Biostatistics, Institute of Public Health, College of Medicine and Health Sciences, University of Gondar, Gondar, Ethiopia, 3 Department of Health System and Policy, Institute of Public Health, College of Medicine and Health Sciences, University of Gondar, Gondar, Ethiopia

* bhamdela@gmail.com

## Abstract

### Background

Short inter-pregnancy interval is a public health concern because it results in adverse perinatal outcomes such as postpartum hemorrhage, anemia, premature birth, low birth weight, and perinatal deaths. Although it is critical to understand the factors that contribute to short inter-pregnancy interval to reduce the risk of these negative outcomes, adequate evidence about the factors in the urban context is lacking. Therefore, we aimed to assess the duration of the inter-pregnancy interval and its predictors among pregnant women in urban South Ethiopia.

### Methods

A community-based retrospective follow-up study was conducted among 2171 pregnant women in five geographically diverse urban settings in South Ethiopia. For the analysis, a Cox gamma shared frailty (random-effect) model was used. Adjusted hazard ratio (AHR) with a 95% CI was used to assess significant predictors. The median hazard ratio (MHR) used to report clustering effect.

### Results

The median duration of the inter-pregnancy interval was 22 months, 95% CI (21, 23), with an inter-quartile range of 14 months. Maternal age ≥30 years [AHR = 0.75, 95% CI: 0.58, 0.97], having no formal education [AHR = 0.60, 95% CI: 0.46, 0.78], contraceptive non-use [AHR = 2.27, 95% CI: 1.94, 2.66], breastfeeding for <24 months [AHR = 4.92, 95% CI: 3.95, 6.12], death of recent child [AHR = 2.90, 95% CI: 1.41, 5.97], plan pregnancy within 24 months [AHR = 1.72, 95% CI: 1.26, 2.35], lack of discussion with husband [AHR = 1.33, 95% CI: 1.10, 1.60] and lack of husband encouragement about pregnancy spacing [AHR =

**Data Availability Statement:** All relevant data are within the manuscript and its Supporting Information files.

**Funding:** The authors received no specific funding for this work.

**Competing interests:** The authors have declared that no competing interests exist.

1.25, 95% CI: 1.05, 1.48] were predictors of short inter-pregnancy interval. Adjusting for predictors, the median increase in the hazard of short inter-pregnancy interval in a cluster with higher short inter-pregnancy interval is 30% [MHR = 1.30, 95% CI: 1.11, 1.43] than lower cluster.

## Conclusions

In the study settings, the duration of the inter-pregnancy interval was shorter than the World Health Organization recommendation. There is a need to improve contraceptive use and breastfeeding duration to maximize the inter-pregnancy interval. Men's involvement in reproductive health services and advocacy for women's reproductive decision-making autonomy are fundamental. The contextual disparities in the inter-pregnancy interval suggests further study and interventions.

## Introduction

Inter-pregnancy interval (IPI), also known as birth to pregnancy interval, is defined as the time elapsed from a live birth to subsequent conception or a woman's last menstrual period (LMP) [1]. The World Health Organization (WHO) recommended at least 24 months between a live birth and subsequent conception, and IPI less than 24 months is generally considered short [1, 2].

Every day in 2017, approximately 810 maternal deaths occur around the world, with 94% occurring in low and middle-income countries [3]. Ethiopia, with a Maternal Mortality Ratio (MMR) of 412 per 100,000 live births, is one of the countries in Sub-Saharan Africa that contributes to unacceptably high level of maternal mortality (295,000) worldwide [3, 4]. Women in developing countries, including Ethiopia, have many more pregnancies and fertility rate than in developed countries [5]. Pregnancies that are many and IPIs that are short and long are public health concerns that have drawn the attention of policymakers and programs due to the negative outcomes associated with them [1]. To be more specific, both short and long IPIs are associated with poor maternal and neonatal outcomes [1, 6]. For instance, IPI <6, <18 and <24 months are associated with an increased risk of prematurity, low birth weight and small for gestational age, which are linked to neonatal death [7–9]. IPI <18 months is associated with an increased risk of adverse maternal outcomes such as anemia, postpartum hemorrhage, and pre-eclampsia, which are associated with maternal death [7, 10]. Longer IPI (>59 months) is associated with pre-eclampsia [11]. By lowering these risks, adequate IPI improves maternal and child survival.

In Ethiopia, contrary to the WHO recommendation, a sizeable proportion of women were not spacing their pregnancies adequately [1, 4]. Still, more than half of second and higher pregnancies occur within a shorter interval than recommended by WHO (at least 24 months) [4]. More specifically, about 47.9% of women in urban areas have a short IPI <24 months [2]. As in many developing countries, the total fertility rate (TFR) in Ethiopia has declined steadily [5], and it has taken 20 years to decline from 5.5 in 2000 to 4.2 in 2019; on average, one child per woman [12]. The current TFR of 4.2 is still high and no more different from the overall TFR in Sub-Saharan Africa (4.6) [5]. A high TFR and a short IPI may result in suffering and poor maternal health conditions in the country.

Previous research has linked short birth intervals to factors such as age [13], maternal education [13–15], residence [2], wealth index [2, 14], sex of the index child [14], perinatal death [15], breastfeeding [2, 14], contraception [2, 14] and parity [13]. However, complex factors such as socioeconomic, demographic, political, cultural, population, and health services have been noted to have an impact on IPI [16]. To the best of our knowledge, despite access to services, the factors contributing to short IPI in urban settings are unclear.

This study was proposed to fill this knowledge gap by examining the effect of clustering using a shared frailty modeling approach in five geographically diverse urban settings. Therefore, this study was aimed to assess the duration of the inter-pregnancy interval and its predictors among pregnant women in urban South Ethiopia. The findings will help to strengthen interventions aimed at optimizing inter-pregnancy intervals, such as family planning programs. It will also help to accelerate maternal and child health-related sustainable development goals.

## Methods

### Study setting

This research was carried out in the Hadiya zone, which is located in the Southern Nations, Nationalities and Peoples Region (SNNPR), Ethiopia. The administrative center of the zone is Hossana town. In Hadiya zone, there is one general hospital, three primary hospitals, 62 health centers, and 311 health posts that provide health services to the community [Hadiya Zone Health Bureau report-Unpublished]. In this study, five urban settings (Hossana, Shone, Gimbichu, Jajura, and Homecho), which consist of a total of eighteen kebeles, were included. Kebele is the lowest administrative unit in Ethiopia. It is a part of a district or sub-district that contains households or a delimited group of people.

### Study design and population

A community-based retrospective follow-up study was carried out in five urban settings in Hadiya zone, South Ethiopia. The study was conducted among pregnant women who had a live birth during their most recent childbirth from July 1, 2014 onwards (i.e., women who had their most recent live birth within the last five years preceding the start date (July 8, 2019) of data collection). All women who met the following eligibility criteria were included: had at least one live birth during the most recent childbirth; had no recent abortion; had no recent stillbirth; could recall the date of recent childbirth or could show an immunization card; and were pregnant at the time of data collection.

### Sample size and data collection techniques

A sample size of 440 was calculated in Epi Info StatCalc version 7.2.2.6 software using the formula for cohort or cross sectional designs, assuming % outcome (short IPI) in the unexposed group (contraceptive non-users) = 65.1%, % outcome (short IPI) in the exposed group (contraceptive users) = 51.5% from the previous study [2], two-sided confidence level = 95%, and power of 80%. We calculated IPI by subtracting nine months of gestational age from the birth interval [1]. However, from July 8, 2019 to December 30, 2019, a total of 2171 pregnant women were identified through house-to-house identification and included in the study. Face-to-face interviews conducted at house-hold level during the identification, using a structured questionnaire.

### Variables and measurements

The outcome variable was the duration of the inter-pregnancy interval. The inter-pregnancy interval was calculated by subtracting the date of the most recent childbirth from the date of

the last menstrual period (IPI in months = date of LMP minus date of recent childbirth). When women were unable to recall the date of their last menstrual period, gestational age was estimated using Ultrasound and then subtracted from the date that a woman had the Ultrasound scan to obtain the date of LMP (date of LMP = date of Ultrasound scan minus the gestational age at the time of the Ultrasound scan) [1]. A follow-up period was defined as the number of months between a live birth and conception or the women's LMP. The event (failure) was defined as the occurrence of pregnancy after a live birth within 24 months (short IPI), whereas the censored (success) was the absence of pregnancy within 24 months (optimal IPI) [1]. The duration of months spent from live birth to subsequent pregnancy, or the woman's LMP, was a time variable. The independent variables included in this study were: 1) sociodemographic and economic, such as religion, ethnicity, marital status, maternal age, sex of child, the number of children, education, occupation, and wealth index. 2) Reproductive characteristics such as parity, age at first childbirth, mode of delivery, and breastfeeding 3) Health-related services such as contraceptive use, antenatal and postnatal care visits, counselling, place of delivery. 4) Decision-making such as decision-making for contraceptive use, discussion with husband about pregnancy spacing and whether the husband encourages wife to space pregnancies or not (Table 1).

The wealth index was measured using household assets for urban residences, which include the following items: the owner of the house; the number of rooms; the material of the roof; the material of the floor; the material of the exterior wall; the source of drinking water; the type of latrine; the type of cooking materials (1 = electricity, 0 = wood/charcoal/biogas/natural gas, etc.), the source of income, and the presence or absence of: cell phone, refrigerator, radio, television, stove, chair, table, watch, modern bed, bicycle, Bajaj (three-wheeled vehicle), motor cycle, car, donkey/horse cart, and bank account. Each item was categorized into two (1 = yes and 0 = no). Based on the World Food Program and WHO recommendations, latrines and water sources were categorized as improved and unimproved facilities [17]. Principal component analysis was done to generate the components. Finally, ranking was done in five categories (lowest, second, middle, fourth, and highest).

## Quality control measures

The questionnaire was developed in English from related literature and the Ethiopia Demographic and Health Survey (EDHS) and translated into the Amharic language. It was pretested in a similar setting (Durame Town). Two days of training were given for data collectors (ten midwives) and supervisors (five public health professionals) on the concept and approaches to the participants. Supervisors closely monitored the data collection process. To minimize recall bias related to recalling the date of the last childbirth, we limited the date of the most recent childbirth to the previous five years. Family members such as the husband, grandparents, and mother-in-law were also involved to recall the date of childbirth. To reduce selection bias, all pregnant women during the study period were included based on predetermined eligibility criteria. Furthermore, Ultrasound was used for women who had difficulty remembering the date of their LMP for a variety of reasons, including contraceptive use and breastfeeding. Epi-data was used to control data entry errors.

## Analysis

The data were entered into Epi-data version 3.1 and analyzed in Stata version 14. Prior to analysis, the data were explored to check outliers and missing values. For continuous variables, descriptive statistics such as mean, median, and standard deviations were calculated. For categorical variables, frequencies and percentages were computed. A complete case analysis was performed for

**Table 1. List of variables, definitions and measurements for the study in urban South Ethiopia, 2019.**

| Variables | Measurements |
|---|---|
| **Inter-pregnancy interval (IPI)** | Time duration from date of live birth to date of woman's last menstrual period in months. It was categorized as event if pregnancy occurred in <24 months (short IPI) or censored if it occurred at ≥24 months (optimal IPI). The categorization was based on World Health Organization recommendation for pregnancy spacing [1]. |
| **Maternal age** | The reported age of a woman at the time of the interview in completed year. It was categorized as 20–24, 25–29 and ≥30 years |
| **Maternal education status** | Education level of a woman at the time of interview. It was categorized as no formal, primary, secondary and higher education |
| **Age at first childbirth** | Reported age at the time that a woman had her first childbirth. It was categorized as <20 and ≥20 years. |
| **Greater number of children by sex** | Whether the family has more female, male or equal number of children by sex, which was categorized as female, male and equal |
| **Parity** | The number of times a woman has given birth, regardless of the outcomes. It was classified as 1, 2, 3 and ≥4 |
| **Number of previous ANC visits** | The number of antenatal care visits a woman made during her previous pregnancy. It was categorized as <4 and ≥4. |
| **Recent number of children** | The total number of children that the family recently has, categorized as 0–1, 2–3 and ≥4. |
| **Past history of Stillbirth** | Whether a woman had a history of giving birth to a baby with no signs of life such as no breathing, no heartbeat, and no movement prior to the most recent delivery. It was answered as 'yes' if present, 'no' otherwise. |
| **Survival status of the recent child** | Whether the most recent child was alive or died, responded as 'yes' if alive, 'no' if died. |
| **Counselled during previous ANC visits** | Whether or not the woman received advice from health care providers about spacing pregnancies during previous antenatal visits, responded by a yes or no response. |
| **Counselled during PNC** | Whether or not the woman received advice from health care providers about spacing pregnancies during the postnatal period, including child immunization visits, was responded by a yes or no response. |
| **Exclusive breastfeeding** | If the mother had only breast milk for her most recent child for up to six months, without any additional food or fluids, except medications, the answer was yes if <6 months and no if ≥6 months. |
| **Total duration of breastfeeding** | Women were asked how long they had breastfed their most recent child until it was discontinued in months, and then the reported number of months categorized as <24 and ≥24. |
| **Decision maker on contraception** | Whether a wife, husband, or both made the decision to use contraception when it was necessary, it was reported as wife alone, husband alone, and jointly (both). |
| **Discussion with husband** | If the wife has discussed or talked with her husband about spacing pregnancies after a recent childbirth, responded as yes or no. |
| **Husband encourages spacing** | Whether a husband encourages his wife to space pregnancies via safe methods of contraception when she requested and/or himself advise her to use responsibly, responded as yes or no. |
| **Modern contraceptive use** | If a woman used any form of modern contraception after recent childbirth, responded as yes or no |
| **Plan to wait until current pregnancy** | Refers to how long couples planned (if they planned) to wait between their most recent childbirth and their current pregnancy in months, categorized as <24 and ≥24 |
| **Desired number of children** | Refers to the number of children that the couples (both husband and wife) wish to have in agreement, categorized as 1–5, ≥ 6 and undecided. If the woman and her husband could not agree on the number of children or had different wishes, it was classified as "undecided." |
| **Mode of delivery for the recent child** | Refers to the process of delivery for the most recent child, and categorized as spontaneous vaginal, cesarean section and instrumental (forceps or vacuum) delivery. |

the missing data. A survival analysis model was fitted since IPI is a time to event variable (from live birth to pregnancy). The Kaplan-Meir or product limit estimator was used to estimate cumulative survival probabilities and compare survival for the predictors. A Log-rank test was used to test the quality of survival between different groups and see if the graphs were significantly different for predictors of IPI. Because the data came from 18 different clusters (kebeles), the clustering effect (between cluster variations) was examined using the frailty variance of theta in the Cox gamma shared frailty model (null model). Kebele was used as a clustering variable.

Variables that showed a statistically significant association with short IPI at P<0.20 in the bivariable Cox gamma shared frailty model were selected for adjustment in the multivariable model. Variables that showed a statistically significant association at P<0.05 and 95% CI for adjusted hazard ratio that did not include 1 in the multivariable Cox gamma shared frailty model were reported as predictors of short IPI. In the adjusted model, interaction for possible effect modification was checked. A model with a better fit was selected by using log-likelihood, Akaike Information Criteria (AIC) and Bayesian Information Criteria (BIC). The results were interpreted using the hazard ratio (HR) as an effect measure. A clustering effect was estimated using the median hazard ratio (MHR).

### Ethics approval and consent to participate

Ethical approval was obtained from the Institutional Review Board (IRB) of the University of Gondar, with registration number: O/V/P/RCS/05/1051/2019. Permission was obtained from regional and local health offices. The study participants were informed about how they were included in the study, the purpose of the study, their rights to withdraw or continue, and the potential benefits and harms of the study. A written consent form was prepared and attached together with the questionnaire to obtain approval from each study participant by signature or fingerprint.

## Results

### Socio-demographic and reproductive characteristics

A total of 2202 pregnant women who fulfill the inclusion criteria in the five urban settings were identified. Of these, 31 women were refused to participate. This corresponds to a response rate of 98.6%. The study included the remaining 2171 pregnant women. The gestational age of the pregnant women at the time of the interview ranges from 12 to 24 weeks. The majority, 2149 (99%), of the pregnant women were in the second trimester of pregnancy. The rest, 22 (1%), were at the end of the first trimester of pregnancy. The mean age of the pregnant women was 27.31 ± 3.44 years. The ages range from 20 to 40 years. The majority, 1142 (54.5%), belong to the age group of 25–29 years. Among the participants, 1936 (89.3%) were Hadiya in ethnicity (**Table 2**).

### Duration of inter-pregnancy interval

The median duration of IPI was 22, 95% CI (21, 23) months, with an inter-quartile range of 14 months. During the follow-up, a total of 1199 (55.2%) pregnancies occurred within a short interval (<24months). The cumulative probability of survival (remaining not pregnant) decreased as the months after live birth increased **(Fig 1)**.

### Predictors of short inter-pregnancy interval

For these clustered data, shared frailty or unobserved heterogeneity due to clustering was checked by fitting a Cox gamma shared frailty model (null model). The Cox gamma shared

**Table 2. Socio-demographic and reproductive characteristics of pregnant women in five urban settings in Hadiya zone, South Ethiopia, 2019.**

| Variables | Frequency (%) |
|---|---|
| **Religion** | |
| Protestant | 1898 (87.4) |
| Orthodox | 115 (5.3) |
| Catholic | 95 (4.4) |
| Muslim | 52 (2.4) |
| Apostolic | 11 (0.5) |
| **Ethnicity** | |
| Hadiya | 1936 (89.2) |
| Kembata/Tembaro | 114 (5.3) |
| Guragie/Siltie/Amhara/Oromo | 121 (5.5) |
| **Marital status** | |
| Married | 2133 (98.2) |
| Unmarried/ Divorced | 38 (1.8) |
| **Woman's main occupation** | |
| Housewife | 1613 (74.3) |
| Employed | 339 (15.7) |
| Merchant/ farmer/daily laborer/waiter | 219 (10) |
| **Husband's main occupation** | |
| Daily laborer | 599 (27.6) |
| Merchant | 554 (25.5) |
| Employed | 663 (30.6) |
| Farmer | 274 (12.6) |
| Driver | 81 (3.7) |
| **Husband's education** | |
| No formal education | 315 (14.6) |
| Primary school (1–8 grades) | 830 (38.4) |
| Secondary school (9–12 grades) | 487 (22.5) |
| Higher education | 532 (24.5) |
| **Gravidity** | |
| 2–3 | 1462 (67.4) |
| ≥ 4 | 709 (32.6) |
| **Place of delivery for recent child** | |
| Home | 369 (17) |
| Health facility | 1802 (83) |

frailty model (null model) with efron (method to handle tied failures) produced a higher Log-likelihood (-8772.261), lower AIC (17544.52) and BIC (17544.52), and was selected as a good fit model for our data. In the Cox gamma shared frailty model, the null model frailty variance of theta equals 0.05, 95% CI (0.01, 0.09), LR test of theta = 0: $X^2$ = 44.15, P<0.001 indicates that there is unobserved heterogeneity or shared frailty, as frailty variance of theta and its 95% CI are greater than zero. That means, women from the same cluster are more or less prone to have short IPI, and assuming dependency (correlation) within a cluster and variation between clusters would yield more reliable estimates of the predictors. In the bivariable Cox gamma shared frailty model, maternal age, maternal education, parity, age at first childbirth, greater number of children by sex, recent number of children, desired number of children, number of previous ANC visits, past history of stillbirth, survival status of recent child, counseling during

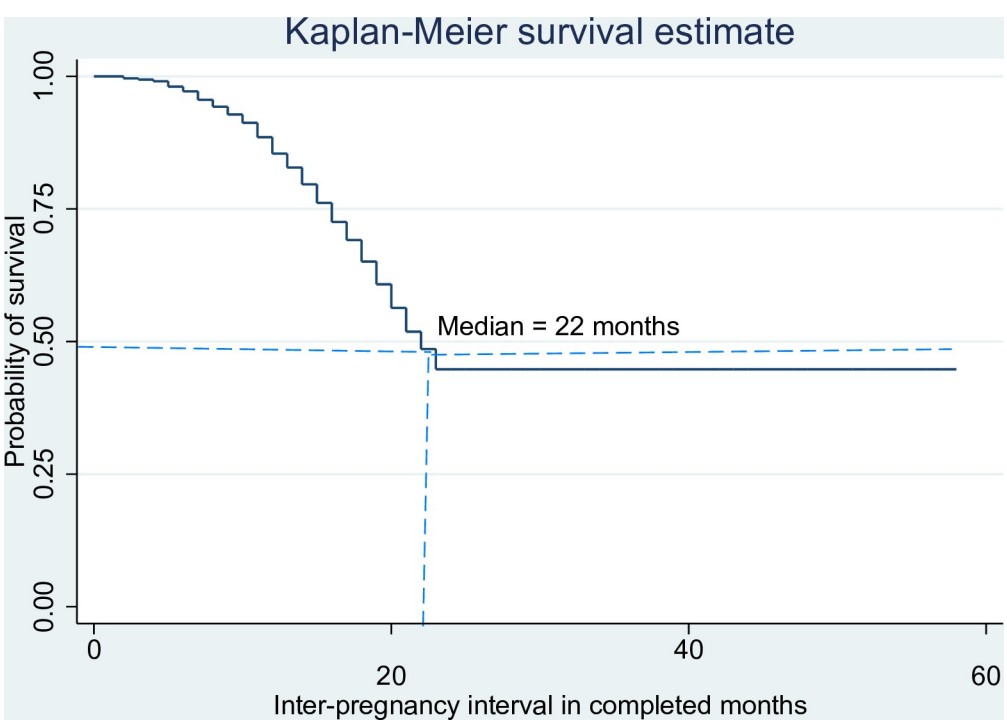

**Fig 1. Cumulative survival time to pregnancy in completed months among pregnant women in urban South Ethiopia, 2019.**

ANC, counseling during PNC, exclusive breastfeeding, total duration of breastfeeding, mode of delivery for the recent child, decision-making for contraception, discussion with husband, husband encouragement for pregnancy spacing, modern contraceptive use, planning pregnancy, and wealth status were significantly associated with short IPI at P<0.20.

In multivariable Cox gamma shared frailty model, maternal age ≥30 years, having no formal education, contraceptive non-use, short duration of breastfeeding, death of recent child, planning pregnancy for <24 months, not discussing with husband about pregnancy spacing and husband not encouraging pregnancy spacing were found to be statistically significant predictors of short IPI, with a 95% confidence level and P<0.05.

Accordingly, women who were 30 years or older were 25% [AHR = 0.75, 95% CI: 0.58, 0.97] less likely to have a pregnancy within a short period of time after giving birth than women who were 20–24 years old. Women with no formal education were 40% [AHR = 0.60, 95% CI: 0.46, 0.78] less likely to have a pregnancy within a short period of time after giving birth than those with a higher education. Women who did not use any modern contraceptive methods after a recent childbirth were twice [AHR = 2.27, 95% CI: 1.94, 2.66] more likely to become pregnant shortly after childbirth than contraceptive users. Women who stopped breastfeeding their most recent child within 24 months were nearly five times more likely to become pregnant in a short period of time [AHR = 4.92, 95% CI: 3.95, 6.12] than those who continued breastfeeding for the recommended duration (>24 months). Women who had lost the most recent child (in the first month of life) were nearly three times [AHR = 2.90, 95% CI: 1.41, 5.97] more likely to have a pregnancy within a short period of time than women whose most recent child was alive. Women who planned to become pregnant within 24 months after a live birth were nearly twice [AHR = 1.72, 95% CI: 1.26, 2.35] more likely to have a pregnancy within a short duration than women who planned for 24 or more months. Women who had

no discussion with their husband about pregnancy spacing were 33% [AHR = 1.33, 95% CI: 1.10, 1.60] more likely to have a pregnancy within a short duration after childbirth than their counterparts. Similarly, women whose husband did not encourage pregnancy spacing were 25% [AHR = 1.25, 95% CI: 1.05, 1.48] more likely to have a pregnancy within a short duration after live birth than women whose husband encouraged pregnancy spacing.

Adjusting for predictors, the median increase in the hazard of short IPI when comparing a woman in a cluster (kebele) with a higher short IPI to a woman in a cluster with a lower short IPI was 30% [MHR = 1.30, 95% CI: 1.11, 1.43] [18] (**Table 3**).

## Discussion

In this study, the median duration of IPI in urban settings was found to be short. Maternal age ≥30 years and having no formal education were protective factors of short IPI. Non-use of modern contraceptive methods, short duration of breastfeeding, death of the recent child, planning pregnancy within a short duration, not having discussion with the husband and husband not encouraging pregnancy spacing were found to increase the chance of having a pregnancy within a short duration after birth.

Despite the attempts made to minimize it, this study might have limitations. This study relied on retrospective follow-up data, so it might have had recall bias. Estimating gestational age using ultrasound and LMP might give different intervals to some extent. Hence, the cumulative survival graph and survival probabilities need to be interpreted taking this into consideration. Despite the limitations, it will provide useful information for planning maternal health services.

The estimated median duration (22 months) of IPI in urban settings was lower than the estimates of Lemo, 24 months [2] and Dabat, 23.6 months [19] districts and the national demographic and health survey, 25.5 months [4], Tanzania, 24.4 months [13] and Bangladesh, 46 months [15]. The variations could be due to differences in estimating IPI using LMP. We calculated IPI based on the date of LMP and the Ultrasound results of pregnant women. For this discussion, IPI from those studies was estimated by subtracting nine months of gestational age from the birth interval. Population characteristics, study settings, and sample size could be the other reasons. The desire for the number of children might be varied in different settings, even within the same country. Individual preferences for rearing children might vary; some people might wish to have children too closely so that they reach the desired number of children within a few years and then go on to their business, such as education and income-generating activities to support their family. The estimated duration of IPI in this study was less than the WHO recommended duration of 24 months [1]. This indicates that the presence of access to health services in urban areas cannot assure the use of modern contraceptive methods to space pregnancies adequately.

In this study, women who were 30 years or older were less likely to have a pregnancy within a short duration after childbirth as compared to those belong to 20–24 years of age. As the ages of women increased, possibly they might have the desired number of children or family size than those belong to an earlier-ages. It could also be due to fertility decline or delays; as women age increases (>30 years), fertility is likely decreased or delayed [20]. On the other hand, the younger age group, especially those who had lower birth order, would have limited experiences of fertility regulation, fewer exposures to health facility visits for maternal health services, and subsequent counseling on the need to space pregnancies. Thus, they might have fewer experiences with pregnancies and childbirth difficulties as well [15, 21]. This finding is consistent with the findings of other studies, which found that older women were less likely to have shorter intervals [13, 22].

**Table 3. Multivariable Cox gamma shared frailty analysis for the predictors of short inter-pregnancy interval among pregnant women in urban South Ethiopia, 2019.**

| Variables | Pregnancy status | | CHR (95% CI) | AHR (95% CI) |
|---|---|---|---|---|
| | Event (<24 months) | Censored (≥24 months) | | |
| **Maternal age** | | | | |
| 20–24 | 218 (60.6) | 142 (39.4) | 1 | 1 |
| 25–29 | 633 (55.4) | 509 (44.6) | 0.88 (0.75, 1.03) | 0.84 (0.69, 1.03) |
| ≥ 30 | 307 (51.9) | 285 (48.1) | 0.77 (0.64, 0.93)** | 0.75 (0.58, 0.97)* |
| **Maternal education status** | | | | |
| No formal education | 209 (50.6) | 204 (49.4) | 0.94 (0.77, 1.16) | 0.60 (0.46, 0.78)*** |
| Primary | 515 (55.9) | 407 (44.1) | 1.08 (0.91, 1.28) | 0.82 (0.66, 1.00) |
| Secondary | 278 (59.8) | 187 (40.2) | 1.21 (1.00, 1.45)* | 1.01 (0.82, 1.25) |
| Higher | 196 (53.1) | 173 (46.9) | 1 | 1 |
| **Age at first childbirth** | | | | |
| < 20 | 283 (54.7) | 234 (45.3) | 1 | 1 |
| ≥ 20 | 890 (55.3) | 718 (44.7) | 1.21 (1.04, 1.39)* | 1.18 (0.98, 1.42) |
| **Greater number of children by sex** | | | | |
| Female | 531 (59.1) | 365 (40.9) | 1 | 1 |
| Male | 460 (54.1) | 391 (45.9) | 0.94 (0.83, 1.06) | 1.03 (0.89, 1.20) |
| Equal | 205 (49.5) | 209 (50.5) | 0.79 (0.67, 0.93)** | 0.83 (0.67, 1.05) |
| **Parity** | | | | |
| 1 | 511 (58) | 370 (42) | 1.32 (1.11, 1.57)** | 0.59 (0.20, 1.74) |
| 2 | 332 (53.4) | 290 (46.6) | 1.15 (0.95, 1.38) | 0.67 (0.28, 1.61) |
| 3 | 162 (57) | 122 (43) | 1.17 (0.95, 1.44) | 0.73 (0.31, 1.74) |
| ≥ 4 | 192 (50.8) | 185 (49.2) | 1 | 1 |
| **Number of previous ANC visits** | | | | |
| <4 | 479 (59.4) | 328 (40.6) | 1.24 (1.07, 1.42)** | 1.06 (0.89, 1.27) |
| ≥ 4 | 574 (51.1) | 549 (48.9) | 1 | 1 |
| **Recent number of children** | | | | |
| 0–1 | 520 (58.2) | 374 (41.8) | 1 | 1 |
| 2–3 | 482 (54.2) | 407 (45.8) | 0.85 (0.75, 0.96)* | 0.97 (0.51, 1.84) |
| ≥ 4 | 187 (50.5) | 183 (49.5) | 0.76 (0.63, 0.89)** | 0.70 (0.24, 2.06) |
| **Past history of Stillbirth** | | | | |
| Yes | 22 (75.9) | 7 (24.1) | 1.76 (1.15, 2.68)** | 1.18 (0.67, 2.03) |
| No | 1174 (54.9) | 964 (45.1) | 1 | 1 |
| **Survival status of the recent child** | | | | |
| Alive | 1172 (54.7) | 970 (45.3) | 1 | 1 |
| Died | 25 (92.6) | 2 (7.4) | 6.28 (4.19, 9.41)*** | 2.90 (1.41, 5.97)** |
| **Counselled during previous ANC visits** | | | | |
| Yes | 790 (50.6) | 770 (49.4) | 1 | 1 |
| No | 401 (67.6) | 192 (32.4) | 1.62 (1.42, 1.84)*** | 1.08 (0.85, 1.37) |
| **Counselled during PNC visits** | | | | |
| Yes | 768 (50.4) | 756 (49.6) | 1 | 1 |
| No | 428 (66.9) | 212 (33.1) | 1.59 (1.40, 1.81)*** | 1.02 (0.81, 1.29) |
| **Exclusive breastfeeding** | | | | |
| Yes | 1027 (53.5) | 892 (46.5) | 1 | 1 |
| No | 161 (68.2) | 75 (31.8) | 1.67 (1.40, 1.96)*** | 0.90 (0.63, 1.29) |
| **Duration of breastfeeding** | | | | |
| <24 months | 1012 (69) | 455 (31) | 4.76 (4.01, 5.66)*** | 4.92 (3.95, 6.12)*** |

(*Continued*)

**Table 3.** (Continued)

| Variables | Pregnancy status | | CHR (95% CI) | AHR (95% CI) |
|---|---|---|---|---|
| | Event (<24 months) | Censored (≥24 months) | | |
| ≥ 24 months | 154 (23.6) | 499 (76.4) | 1 | 1 |
| **Decision maker for contraception** | | | | |
| Husband | 305 (66.9) | 151 (33.1) | 1 | 1 |
| Wife | 92 (53.5) | 80 (46.5) | 0.78 (0.61, 1.00) | 1.31 (0.96, 1.79) |
| Jointly | 795 (51.9) | 738 (48.1) | 0.67 (0.58, 0.77)*** | 1.07 (0.88, 1.29) |
| **Discussion with husband** | | | | |
| Yes | 740 (48.5) | 786 (51.5) | 1 | 1 |
| No | 442 (71.4) | 177 (28.6) | 2.05 (1.81, 2.32)*** | 1.33 (1.10, 1.60)** |
| **Husband encourages spacing** | | | | |
| Yes | 767 (50.4) | 755 (49.6) | 1 | 1 |
| No | 412 (68.1) | 193 (31.9) | 1.72 (1.51, 1.95)*** | 1.25 (1.05, 1.48)* |
| **Modern contraceptive use** | | | | |
| Yes | 420 (39.7) | 639 (60.3) | 1 | 1 |
| No | 778 (70) | 333 (30) | 2.76 (2.44, 3.12)*** | 2.27 (1.94, 2.66)*** |
| **Plan to wait until current pregnancy** | | | | |
| <24 months | 60 (82.2) | 13 (17.8) | 2.56 (1.97, 3.34)*** | 1.72 (1.26, 2.35)** |
| ≥ 24 months | 1035 (54.2) | 876 (45.8) | 1 | 1 |
| **Desired number of children** | | | | |
| 1–5 | 348 (57) | 263 (43) | 1 | 1 |
| ≥ 6 | 656 (54.8) | 542 (45.2) | 0.95 (0.83, 1.09) | 0.87 (0.74, 1.03) |
| Undecided | 195 (53.9) | 167 (46.1) | 0.88 (0.73, 1.06) | 0.84 (0.66, 1.07) |
| **Mode of delivery for recent child** | | | | |
| Spontaneous vaginal delivery | 1116 (55.8) | 884 (44.2) | 1 | 1 |
| Cesarean-section | 51 (46.8) | 58 (53.2) | 0.79 (0.60, 1.05) | 0.77 (0.55, 1.07) |
| Instrumental (forceps and vacuum) | 31 (50.8) | 30 (49.2) | 0.95 (0.66, 1.38) | 1.29 (0.82, 2.03) |
| **Wealth status** | | | | |
| Lowest | 262 (60.9) | 168 (39.1) | 1 | 1 |
| Second | 234 (53.8) | 201 (46.2) | 0.90 (0.75, 1.07) | 0.83 (0.66, 1.03) |
| Middle | 249 (58.5) | 177 (41.5) | 1.01 (0.84, 1.21) | 0.95 (0.76, 1.19) |
| Fourth | 228 (52.7) | 205 (47.3) | 0.89 (0.74, 1.07) | 0.88 (0.71, 1.11) |
| Highest | 216 (50.1) | 215 (49.9) | 0.86 (0.71, 1.05) | 0.94 (0.74, 1.19) |
| **Exclusive breastfeeding* Contraception** | | | | |
| No contraception*no exclusive breastfeeding | | | | 1.82 (1.17, 2.82)** |
| **Plan to wait until current pregnancy*survival status of the recent child** | | | | |
| <24 months*died | | | | 5.30 (1.06, 26.54)* |
| **Theta** | | | | 0.077 (0.011, 0.142)*** |
| **MHR** | | | | 1.30 (1.11, 1.43)*** |
| **LR test of theta = 0** | | | | |
| Chibar2(01) | | | | 41.64 |
| Prob-hibar2 | | | | <0.001 |

Keys: *** = P<0.001,

** = P<0.01,

* = P<0.05,  = P<0.20. AHR: Adjusted Hazard Ratio. CHR: Crude Hazard Ratio. CI: Confidence Interval. MHR: Median Hazard Ratio. LR: Likelihood Ratio.

1 = reference category.

Women with no formal education were less likely to have a pregnancy within a short period of time after birth than women with a higher level of education. Previous studies in Tanzania [13], Ethiopia [14], and Iran [21] found that less educated women had shorter birth intervals. However, the finding is consistent with that of Bangladesh [15] and Korea [23], which found that women with a higher level of education were more likely to have a short birth interval. Higher educated women might have a better employment opportunities, hence might breast-feed less frequently due to lower contact time or lack of breaks, far away work places, full-time employment, and inflexible working time that contribute to the fast return of fertility and increased risk of getting pregnant [24–26]. On the other hand, they might have further education plans so that they might stick to the schedule and wish to have the desired number of children in a few years and give other breast milk substitutes instead of breastfeeding for a longer duration [27].

Women who did not use modern contraceptive methods after a recent childbirth were more likely to become pregnant within a short period of time than contraceptive users, which is consistent with other studies [2, 19, 21, 28, 29]. Contraception is the main tool to achieve optimal pregnancy interval, but can be affected by various factors that health service programs have to address, including myths and misconceptions (mainly fear of infertility) [30].

Women who breastfed their most recent child for a shorter period of time (24 months) had a pregnancy sooner than those who breastfed for the optimal period of time ($\leq$24 months). The finding is consistent with previous research in South Ethiopia [2], Bangladesh [15], Iran [21], and North Ethiopia [28]. This suggests breastfeeding has a positive impact on the duration of IPI, and encouraging women to prolong breastfeeding duration is beneficial. When a woman exclusively breastfeeds her child, it can delay the return of menses and promote pregnancy spacing [31, 32]. Extending breastfeeding after exclusive breastfeeding helps to space pregnancies while also increasing child survival [33]. Although personal, cultural, social, and environmental factors can influence breastfeeding frequency and duration [34], Ethiopia has a good culture of breastfeeding for a longer duration that needs to be promoted for further achievements in increasing pregnancy interval [35].

Modern contraception and exclusive breastfeeding have a positive (agonistic) interaction. Women who did not use modern contraception methods and did not exclusively breastfeed had a higher risk of becoming pregnant soon after birth [AHR = 1.82, 95% CI: 1.17, 2.82]. This indicates that modern contraceptive use and exclusive breastfeeding are highly effective strategies for spacing pregnancies.

Women who planned to become pregnant within 24 months after their recent childbirth were more likely to have a pregnancy within a short duration than those who planned for 24 or more months. This suggests that simply having a plan may not be sufficient; rather, how long is important in planning the next pregnancy. Those who had planned their next pregnancy for a longer period of time might have had their own reasons. As a result, they might have used contraception more consistently to achieve their goals than those who planned for a shorter period of time [29].

Women who lost their recent child (within the first months of life in this study) were more likely to have a short duration of inter-pregnancy interval than their counterparts. Women who lost their last child won't breastfeed to benefit from lactation amenorrhea. Thus, they were more likely to have menses return soon and more likely to get pregnant unless contraceptive methods were used. Those women might also be encouraged by significant others to replace the lost child within a short duration of time, which might help them to recover from psychosocial impacts and forget the lost child. The finding was supported by the study conducted in North-West Ethiopia [19], in which the death of the index child contributed to the occurrence of births within a short period of time.

There was a positive interaction between planning pregnancy within a short duration after a live birth and the death of the index child [AHR = 5.30, 95% CI: 1.06, 26.54]. Women who had already planned to become pregnant soon after childbirth and, if their child died, might have a strong desire to replace the lost child soon.

Women who did not discuss about pregnancy spacing following recent childbirth had a pregnancy within a short duration compared to their counterparts. It is obvious that pregnancy and childbirth need a joint decision of couples/relatives. Women who had a discussion with their husband might have used postpartum contraception in order to space pregnancy [36, 37].

Women whose husbands did not encourage pregnancy spacing were more likely to have a pregnancy in a short period of time than women whose husbands did encourage pregnancy spacing. This suggests that the decision to space the pregnancy is made by the husband. Without the consent of her husband, a woman might not use any form of contraception to space pregnancies [37]. In Ethiopia, preference for the number of children might vary between couples, and the husband might have a greater wish for additional children than the woman [38]. It is also common for husbands to make decisions about most household issues and maternal health services. A husband's involvement may be an important component of maternal health services, including family planning methods [29, 38].

The median hazard ratio indicated that there is variation in inter-pregnancy interval duration due to variation among clusters, highlighting the need for further investigation of contextual factors to implement evidence-based interventions in the urban community. Considering the limitations mentioned above, the findings of this study could be generalized to similar urban settings with similar populations.

## Conclusions

Although maternal health services such as modern contraceptive methods and information are available in urban areas, this study found that more than half of women in urban areas have a shorter duration of IPI than the WHO recommendation. The short duration of IPI was associated with modifiable factors such as non-use of modern contraceptive methods, short duration of breastfeeding, planning pregnancy within a short duration after childbirth, and lack of decision-making autonomy of women to use maternal health services. Therefore, contraceptive utilization, duration of breastfeeding, and planning time of pregnancy have to be improved. Men's involvement in reproductive health services and advocating reproductive decision-making autonomy of women are also fundamental, as decisions to use maternal health services like contraception are usually made by men. Women with higher education levels have to give emphasis to optimal pregnancy spacing. Taking into account contextual differences in inter-pregnancy interval when providing maternal and child health services may help to reduce the risk of short inter-pregnancy interval in the community.

## Supporting information

**S1 Fig. Kaplan-Meir survival graphs for the predictors of short IPI.**
(DOC)

**S1 Table. Log-rank test for the potential predictors (unadjusted) of short IPI.**
(DOCX)

**S1 Appendix. R code to calculate median hazard ratio.**
(DOCX)

**S1 File. Dataset.**
(DTA)

## Acknowledgments

We are very much thankful for study participants, data collectors and supervisors for their contributions.

## Author Contributions

**Conceptualization:** Belayneh Hamdela Jena, Gashaw Andargie Biks, Yigzaw Kebede Gete, Kassahun Alemu Gelaye.

**Data curation:** Belayneh Hamdela Jena, Gashaw Andargie Biks, Yigzaw Kebede Gete, Kassahun Alemu Gelaye.

**Formal analysis:** Belayneh Hamdela Jena, Gashaw Andargie Biks, Yigzaw Kebede Gete, Kassahun Alemu Gelaye.

**Funding acquisition:** Belayneh Hamdela Jena, Gashaw Andargie Biks, Yigzaw Kebede Gete, Kassahun Alemu Gelaye.

**Investigation:** Belayneh Hamdela Jena, Gashaw Andargie Biks, Yigzaw Kebede Gete, Kassahun Alemu Gelaye.

**Methodology:** Belayneh Hamdela Jena, Gashaw Andargie Biks, Yigzaw Kebede Gete, Kassahun Alemu Gelaye.

**Project administration:** Belayneh Hamdela Jena, Gashaw Andargie Biks, Yigzaw Kebede Gete, Kassahun Alemu Gelaye.

**Resources:** Belayneh Hamdela Jena, Gashaw Andargie Biks, Yigzaw Kebede Gete, Kassahun Alemu Gelaye.

**Software:** Belayneh Hamdela Jena, Gashaw Andargie Biks, Yigzaw Kebede Gete, Kassahun Alemu Gelaye.

**Supervision:** Belayneh Hamdela Jena, Gashaw Andargie Biks, Yigzaw Kebede Gete, Kassahun Alemu Gelaye.

**Validation:** Belayneh Hamdela Jena, Gashaw Andargie Biks, Yigzaw Kebede Gete, Kassahun Alemu Gelaye.

**Visualization:** Belayneh Hamdela Jena, Gashaw Andargie Biks, Yigzaw Kebede Gete, Kassahun Alemu Gelaye.

**Writing – original draft:** Belayneh Hamdela Jena, Gashaw Andargie Biks, Yigzaw Kebede Gete, Kassahun Alemu Gelaye.

**Writing – review & editing:** Belayneh Hamdela Jena, Gashaw Andargie Biks, Yigzaw Kebede Gete, Kassahun Alemu Gelaye.

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
