## [Decision Letter · Decision Letter 0]

13 Jan 2022

PONE-D-21-12390Duration of inter-pregnancy interval and its predictors among pregnant women in urban South Ethiopia: Weibull inverse-Gaussian shared frailty modelingPLOS ONE

Dear Dr. Jena,

Thank you for submitting your manuscript to PLOS ONE. After careful consideration, we feel that it has merit but does not fully meet PLOS ONE’s publication criteria as it currently stands. Therefore, we invite you to submit a revised version of the manuscript that addresses the points raised during the review process.

The manuscript has been evaluated by two reviewers, and their comments are available below.<o:p></o:p>

The reviewers have raised a number of concerns that need attention. They request additional information on methodological aspects of the study (such as the study setting and the rationale for some of the approaches used), and revisions to the statistical analyses<o:p></o:p>

Could you please revise the manuscript to carefully address the concerns raised?<o:p></o:p>

We look forward to receiving your revised manuscript.

Kind regards,

Thomas Phillips, PhD

Staff Editor

PLOS ONE

Journal Requirements:

2. Please provide a sample size and power calculation in the Methods, or discuss the reasons for not performing one before study initiation.

3. Thank you for stating the following financial disclosure: "The funders had no role in study design, data collection and analysis, decision to publish, or preparation of the manuscript."

"We would like to thank University of Gondar and Wachemo University for financial support.  We are very much thankful for study participants, data collectors and supervisors for their contributions."

"The funders had no role in study design, data collection and analysis, decision to publish, or preparation of the manuscript"

Reviewers' comments:

Reviewer's Responses to Questions

**Comments to the Author**

1. Is the manuscript technically sound, and do the data support the conclusions?

Reviewer #1: Yes

Reviewer #2: Yes

2. Has the statistical analysis been performed appropriately and rigorously? 

Reviewer #1: Yes

Reviewer #2: I Don't Know

3. Have the authors made all data underlying the findings in their manuscript fully available?

Reviewer #1: Yes

Reviewer #2: Yes

4. Is the manuscript presented in an intelligible fashion and written in standard English?

Reviewer #1: No

Reviewer #2: No

5. Review Comments to the Author

Reviewer #1: The article presents original research. As far I could verify, the results are not published elsewhere. It addresses a significant problem of children, women and community as women in the described community are responsible for both indoor and outdoor responsibilities. Methods and materials are well described and are consistent with the type of the study. The authors used advanced analysis appropriate for data nature to obtain correct information from the data. However, before considering publication the paper will be benefited from the following comments.

1. Pleas add cluster effect analysis

2. study design, population and quality control (better to add this for title population since the paragraph contain these)

3. why you excluded those gave birth before 15 years. Your study population may be reproductive age women, however, you can interview these women but they may have births before 15 years. which may be the potential factor not to use contraceptive and at all to had no power to decide on their reproductive health.

4. starting from line 153 to paragraph before quality control is part of result but not analysis part.

5. table 2 is part of result but not the method part, so please move into result section.

6. first what is the importance of checking PH assumption as such correlated (have common shared value at cluster ) level. rather it is good to check whether frailty is significant or none significant. if frailty is not significant PH ascription test is ideal to fit cox , stratified cox or parametric model. if frailty is significant cox frailty model is ideal. second to consider parametric model what exploratory analysis do you have conducted to support your justified model (to see this please demonstrate base line hazard (h(to) ) distribution and other parametric assumptions. why you are limited on only (Exponential, Gompertz and Weibull), what about at least log normal and loglogistic, and generalised gamma since ststa support these also.

7. what variable selection method do use (authomethod method or purposive method, if it was purposive method 20 to 25% not 5% for variable selection or specify your reference) ???? 0.25 or 0.2 from bivariable to multivariable model. (or log rank test 0.05)

8. Quality control measures (it should come before analysis)

9. From line 191 to 193 (It is part of analysis but not quality control measure)

10. In your method section you have stated (line 171-173) Bivariable weibull inversgaussian shared frailty analysis before Multivariable analysis but here in the result section from line 219 to 228 you have stated log rank test as a selection of variables before multivariable analysis. so please make consistent and if you had used log rank test please express your method for continuous predictors.

11. you can put this (table 4) as supporting document

12. I have seen your interpretation in your document 1. do you think the interpretation of crude and adjusted hazard ratio interpretation similar? I think this interpretation will work for crude one not for adjusted HR. 2. you had also adjust for clusters but your interpretation did not show this and 3. I recommend to conduct median HR to conduct for the cluster effect.

13. in the method section you had mentioned that you consider effect modification but I haven't seen in result part and your result like Survival status of the recent child and Plan to wait until current pregnancy invite interaction analysis consideration since there is beg CHR and AHR discrepancy.

14. In all document please merge citations (that have more than one citations)

15. Line 296 to 298 as far as you have adjusted for Decision maker for contraception and discussion with husband, as well as wealth how do you consider these as confounder or possible reason.

16. Line 310 to 311 need reference

17. Line 313 to 314, is it study based? If it is please cite.

18. Line 318 to 319 what is your base to consider interaction (your statistical or biological or clinical mechanism), what about other variable interactions, since your CHR and AHR for Survival status of the recent child, Plan to wait until current pregnancy ... show more than 15% crude adjusted discrepancy to consider interaction or confounding.

Reviewer #2: GENERALCOMMENTS: There are several grammatical errors and wrong use of tenses hat need to be corrected in order to make the text more understandable.

METHODS: The a118-119 authors should explain what kebeles are for better understanding. What informed the choice of the 5 urban settings selected for the study? In page 6, lines who performed the ultrasound scans, where and when was this done? The gestational ages of the women should have been added for clarity. In page 6,lines 119=120 it is not clear how the interpregnancy interval was determined from the USS. What about those who were in advanced pregnancy when ultrasonography cannot date pregnancy accurately and did not remember their LMP. In page 9,line 140-the authors should explain what baja is.

6. PLOS authors have the option to publish the peer review history of their article (what does this mean?). If published, this will include your full peer review and any attached files.

Reviewer #1: **Yes: **Reta Dewau

Reviewer #2: No

---

## [Author Response · Author response to Decision Letter 0]

10 Feb 2022

Editor’s comments:

Journal Requirements:

Authors’ response: Dear editor, thank you again for your time and contribution. We looked at the PLOS ONE style requirements to edit the whole manuscript and revised accordingly. 

2. Please provide a sample size and power calculation in the Methods, or discuss the reasons for not performing one before study initiation.

Authors’ response: Thank you dear editor, we included the sample size and power calculation in methods section. 

3. Thank you for stating the following financial disclosure: "The funders had no role in study design, data collection and analysis, decision to publish, or preparation of the manuscript."

Authors’ response: Dear editor, we have revised the financial disclosure statements. We revised as “The authors received no specific funding for this work.”

"We would like to thank University of Gondar and Wachemo University for financial support. We are very much thankful for study participants, data collectors and supervisors for their contributions."

"The funders had no role in study design, data collection and analysis, decision to publish, or preparation of the manuscript"

Authors’ response: Dear editor, we have removed funding-related text from the manuscript. We included this in the revised cover letter.

Authors’ response: Dear editor, we have uploaded dataset as supporting information (S1 File. Dataset). 

Authors’ response: Thank you dear editor, the corresponding author’s ORCID ID was created. 

Authors’ response: Dear editor, thank you for giving useful link to edit the supporting information. The caption for supportive information files was added in the manuscript, and also inside each uploaded supportive documents. We also used Preflight Analysis and Conversion Engine (PACE) digital diagnostic tool to revise figure inside the manuscript.

Reviewers’ comments

Reviewer #1

The article presents original research. As far I could verify, the results are not published elsewhere. It addresses a significant problem of children, women and community as women in the described community are responsible for both indoor and outdoor responsibilities. Methods and materials are well described and are consistent with the type of the study. The authors used advanced analysis appropriate for data nature to obtain correct information from the data. However, before considering publication the paper will be benefited from the following comments.

1. Pleas add cluster effect analysis

Authors’ response: Dear reviewer, thank you for suggesting us to include a newly introduced effect measure (the median hazard ratio) to understand the magnitude of clustering effect. Reporting cluster effect, rather than reporting only the variance and individual characteristics, is useful to understand the importance of cluster for evidence-based interventions.

Thus, we added cluster effect analysis according to your recommendation. We included this in methods section (in the analysis) and result section. Kindly, see page number 12 and 21-22.

2. study design, population and quality control (better to add this for title population since the paragraph contain these)

Authors’ response: Dear reviewer, we have revised this section as ‘study design and population’. The quality control contents were moved to quality control section.

3. why you excluded those gave birth before 15 years. Your study population may be reproductive age women, however, you can interview these women but they may have births before 15 years. which may be the potential factor not to use contraceptive and at all to had no power to decide on their reproductive health.

Authors’ response: Dear reviewer, thank you for the comment. We did not exclude those with age below 15 years. In our study, we included those women who had at least one live birth before current pregnancy, irrespective of their age at first childbirth. We just categorized ‘age at first childbirth’ (15-19 and ≥20 years) after data collection, for the analysis purpose. That means, we did not get a woman whose age at first childbirth was less than 15 years. We revised it as <20 and ≥20 years, this may give more sense.

4. starting from line 153 to paragraph before quality control is part of result but not analysis part.

Authors’ response: Thank you dear reviewer, we moved this to the result section. 

5. table 2 is part of result but not the method part, so please move into result section.

Authors’ response: Thank you dear reviewer, we removed table 2 which was about Cox proportional hazard assumption. As you said below in comment number 6 “since frailty is significant, testing Cox proportional hazard assumption was not relevant”. Thus, in the revised manuscript we removed it.

6. first what is the importance of checking PH assumption as such correlated (have common shared value at cluster ) level. rather it is good to check whether frailty is significant or none significant. if frailty is not significant PH ascription test is ideal to fit cox , stratified cox or parametric model. if frailty is significant cox frailty model is ideal. second to consider parametric model what exploratory analysis do you have conducted to support your justified model (to see this please demonstrate base line hazard (h(to) ) distribution and other parametric assumptions. why you are limited on only (Exponential, Gompertz and Weibull), what about at least log normal and loglogistic, and generalised gamma since ststa support these also.

Authors’ response: Dear reviewer, thank you very much for suggesting us to follow the ideal approach for the survival analysis. We made significant revision in the analysis considering your constructive comment. Of course, in previous manuscript, our approach was parametric (Weibull inverse Gaussian shared frailty analysis). We choose parametric because Cox proportional assumption was violated. Then, we checked presence of frailty (with parametric approach), it was significant and Weibull inverse Gaussian shared frailty analysis was done. Weibull, Gompertz and exponential distributions were selected because they give hazard ratio. But other distributions (log normal, log logistic and generalized gamma) do not give hazard ratio. Rather they give rate ratio which makes the interpretation difficult. Since our aim is to report effect size using hazard ratio we compared the above three distributions and using Weibull found to be good fit model. 

 However, in this revised manuscript, we accepted your suggestion to consider a semi-parametric (Cox gamma shared frailty analysis) which is an ideal approach. As you said, in the revised manuscript we first checked whether the frailty is significant or not, using semi-parametric (Cox gamma) frailty. It was significant (LR test of frailty variance was significant (P<0.001)). The null model frailty variance itself was greater than zero (0.0462576, 95% CI (0.007703, 0.084812)). Thus, shared frailty analysis is appropriate. Testing Cox proportional hazard assumption was not relevant once frailty is significant. So we removed the table that represented cox proportional hazard assumption. Therefore, the analysis in the revised manuscript was according to your suggestion (Cox gamma shared frailty). Thank you again for sharing your experience.

7. what variable selection method do use (authomethod method or purposive method, if it was purposive method 20 to 25% not 5% for variable selection or specify your reference) ???? 0.25 or 0.2 from bivariable to multivariable model. (or log rank test 0.05)

Authors’ response: Thank you dear reviewer for the constructive comment. For mixed effect models, including frailty, it is customary to use P vales such as 0.20 and 0.25 to select variables from bivariable to multivariable model rather than just using 0.05, as this (0.05) cutoff results in loss of important variables that were confounded by others. After your useful comment, we revised the analysis and used P<0.20 as a cutoff to select variables from bivariable to multivariable model. This helps to include as many predictor variables as possible for the adjustment. Of course, it is also possible to use the log-rank test but it did not give a hazard ratio to understand the strength of association. However, we used log-rank test to test the equality of survival across the different categories of a predictor variable. This was reported in supporting information file (S1 Table). 

8. Quality control measures (it should come before analysis)

Authors’ response: Thank you dear reviewer we moved it before analysis section.

9. From line 191 to 193 (It is part of analysis but not quality control measure)

Authors’ response: Thank you dear reviewer we moved it from the quality control measure to the analysis section.

10. In your method section you have stated (line 171-173) Bivariable weibull inversgaussian shared frailty analysis before Multivariable analysis but here in the result section from line 219 to 228 you have stated log rank test as a selection of variables before multivariable analysis. so please make consistent and if you had used log rank test please express your method for continuous predictors.

Authors’ response: Thank you dear reviewer we revised this inside manuscript. We used bivariable Cox gamma shared frailty model to select variables for multivariable model, using P<0.20, as we responded for comment number 6 and 7 above. Now, in the revised manuscript, log-rank test was used only to test equality of survival across different categories of a variable, as reported in supporting information (S1 Table). Continuous variables were categorized or recoded before log-rank test.

11. you can put this (table 4) as supporting document

Authors’ response: Dear reviewer we put it as a supporting information (S1 Table).

12. I have seen your interpretation in your document 1. do you think the interpretation of crude and adjusted hazard ratio interpretation similar? I think this interpretation will work for crude one not for adjusted HR. 2. you had also adjust for clusters but your interpretation did not show this and 3. I recommend to conduct median HR to conduct for the cluster effect.

Authors’ response: the interpretation of hazard ratio in crude and adjusted model is similar, except the difference in the estimation. In the adjusted model, the hazard ratio is adjusted for other covariates. So only the estimates vary in the crude and adjusted model. The interpretation is similar with other effect measure, like relative risk. Both in crude and adjusted models, HR>1 indicate increased risk; HR<1 indicates decreased risk or protective effect of exposure, and HR=1 indicates no difference or no effect. Of course, interpretation of hazard ratio in parametric model is different (as it take accelerated failure time form) from a semi-parametric one. We tried to make the interpretation of hazard ratio easier to understand, especially for those readers outside the field. For example, the interpretation “the hazard of short inter-pregnancy interval was 2 time higher for women who did not use modern contraceptive methods than contraceptive users” may be difficult (‘the hazard of’) to understand by readers outside the field. This can be simplified as “Women who did not use modern contraceptive methods were 2 times more likely to become pregnant within short duration after childbirth than contraceptive users. 

Regarding adjustment for cluster, we were just to say considering for clustering in univariable and multivariable models. We removed the phrase ‘adjusting for clustering’ from the text, it may not be important to report once we consider cluster effect analysis. 

We conducted a median hazard ratio for the cluster effect. In this analysis, we reported both null model and full (adjusted) model median hazard ratios with their interpretations (kindly, see page number 21-22). Thank you again for encouraging us to consider this new effect measure for clustered data.

13. in the method section you had mentioned that you consider effect modification but I haven't seen in result part and your result like Survival status of the recent child and Plan to wait until current pregnancy invite interaction analysis consideration since there is beg CHR and AHR discrepancy.

Authors’ response: Thank you dear reviewer for the constructive comment. Now we considered those variables for the interaction effect (Table 3). Also, we checked the other variables as well.

14. In all document please merge citations (that have more than one citations)

Authors’ response: Thank you dear reviewer now we merged the citations.

15. Line 296 to 298 as far as you have adjusted for Decision maker for contraception and discussion with husband, as well as wealth how do you consider these as confounder or possible reason.

Authors’ response: Thank you dear reviewer for the input. Now we revised in the text. As you said, we already adjusted for those variables so no need to use them as possible reason. Thus, removed from the text.

16. Line 310 to 311 need reference

Authors’ response: Dear reviewer now we added the reference.

17. Line 313 to 314, is it study based? If it is please cite.

Authors’ response: Dear reviewer now we included the citation.

18. Line 318 to 319 what is your base to consider interaction (your statistical or biological or clinical mechanism), what about other variable interactions, since your CHR and AHR for Survival status of the recent child, Plan to wait until current pregnancy ... show more than 15% crude adjusted discrepancy to consider interaction or confounding.

Authors’ response: Dear reviewer, thank for suggesting us to check other variables for possible interactions. Our base to consider the interaction was our theoretical knowledge. For example, breast feeding has a potential to delay ovulation due to the effect of prolactin hormone. Delay in ovulation reduce the chance of getting pregnancy. Hormonal contraceptive methods also have similar mechanism of action (delay ovulation), and then prevent occurrence of pregnancy. These hormonal contraceptive methods are a synthetic hormones which are similar with female hormones (progestin and estrogen). Therefore, breast feeding and at the same time using hormonal contraceptive methods might have synergetic effect (effect modification). Of course, effect modification is not a bias. Reporting the independent effect (by avoiding interaction effect) is useful to identify the independent effect of an exposure variables on the outcome. The other is statistical interaction which might occur due to the nature of data or the effect measure used. It may not necessarily have clinical importance (plausibility) but may affect the parameter estimates and bias the relationship between exposure and outcome, as the value of one covariate depends on the value of the other covariate. Thus, we checked for statistical interaction by fitting all covariates in the adjusted model one by one. If the interaction is significant, we retain the interacted variables in the model. If not significant, we did not consider the variables in the model. In this regard, the only variables found to be significant for the interaction were ‘exclusive breast feeding with contraceptive use’ (biological or clinical interaction), and ‘planning pregnancy within short duration with the survival status (death) of the last child’ (statistical interaction). Interaction between ‘planning pregnancy within short duration and death of last child’ could be statistical because it has no biological or clinical mechanisms rather it might be due to the fact that those women who had a plan to give next child soon after the preceding birth might be provoked by the death of the preceding child. 

In addition to theoretical knowledge, as you said, the gap between crude and adjusted estimates could be a base to invite interaction or confounding. 

Again, thank you for this vital contribution to improve the quality of the analysis.

Reviewer #2 

GENERALCOMMENTS: There are several grammatical errors and wrong use of tenses that need to be corrected in order to make the text more understandable.

Authors’ response: Thank you dear reviewer we tried to correct grammatical error and use of tenses as much as possible in the revised manuscript. 

Reviewer comment: METHODS: The a118-119 authors should explain what kebeles are for better understanding. 

Authors’ response: Dear reviewer now we explained what the kebele is in the methods section (under study setting on page 5). Kebele is the lowest administrative unit in Ethiopia. It is just a part of district or sub-district. The administration system in Ethiopia is as follow: Federal Region Zone District Kebele. Next to kebele is the household or families/person.

Reviewer comment: What informed the choice of the 5 urban settings selected for the study? 

Authors’ response: Dear reviewer, the 5 urban settings were selected for the study for the following reasons. Firstly, as we have mentioned in the ‘introduction’ there is scarcity of evidence about factors contributing to short inter-pregnancy interval in the urban areas even though access for maternal health services are available. Secondly, the 5 urban areas are the only ‘town administrations’ in the zone. Thus, we included all of them. Thirdly, in these urban areas there are hospitals with Ultrasound services. For those women who unable to recall date of last menstrual period, we use this opportunity to estimate gestational age and subsequently calculate date of last menstrual period and inter-pregnancy interval rather than excluding those who unable to recall the dates of last menstrual period. This helps to reduce selection bias related to recalling date of last menstrual period. Fourthly, others are semi-urban which are not that much different from the rural and no such ultrasound facilities to estimate gestational ages as many women in rural areas has difficulty in recalling the dates of last menstrual period.

Reviewer comment: In page 6, lines who performed the ultrasound scans, where and when was this done? 

Authors’ response: Dear reviewer, thank you for the constructive comments. In our setting, it is not feasible to us to scan every woman by Ultrasound. Thus, the last menstrual period recall is mainly used. In this study, the Ultrasound scan was done at the hospitals of each town administration for free of cost (after communication with hospital administration Ultrasound service was allowed for those who could not be able to recall the date of last menstrual period). The Ultrasound scan was performed by those who already give routine Ultrasound scan in the hospitals (radiologists and trained medical Doctors). The Ultrasound scan was done (for those unable to recall LMP) at the end of first trimester and just at second trimester of pregnancy. We included pregnant women who were at the end of first trimester and at second trimester. We did so for two reasons; firstly, it is costly to us to test pregnancy in the first trimester (laboratory-based test). In first trimester, some women even may not know whether they are pregnant. Thus, we included women whose pregnancy was already confirmed. Secondly, women in the advanced pregnancy (third trimester) might have difficulty in recalling the date of last menstrual period, and the Ultrasound is less accurate. Hence pregnant women at third trimester were not included.

Reviewer comment: The gestational ages of the women should have been added for clarity. 

Authors’ response: Dear reviewer, we included the gestational age of women (gestational age at the time of interview in weeks) in the revised manuscript. Kindly see the text on page 13.

Reviewer comment: In page 6,lines 119=120 it is not clear how the interpregnancy interval was determined from the USS. 

Authors’ response: Thank you dear reviewer now we clarified it by including the formula that the inter-pregnancy interval was calculated. 

To calculate inter-pregnancy interval, we need to have two time points; date of last childbirth and date of last menstrual period. If we have these two dates, we can simply subtract date of last childbirth from the date of last menstrual period (i.e. inter-pregnancy interval = date of last menstrual period minus date of last childbirth). This gives inter-pregnancy interval in complete months. If date of last menstrual period is not known, we can calculate it from the gestational age which is determined by ultrasound. This can be done by subtracting the gestational age from the date that a woman was seen by the ultrasound; i.e. date of last menstrual period=date of the ultrasound scan minus the gestational age at the time of Ultrasound scan. This clarification remarks were reported in the revised manuscript. Kindly, see page number 6.

Reviewer comment: What about those who were in advanced pregnancy when ultrasonography cannot date pregnancy accurately and did not remember their LMP. 

Authors’ response: Dear reviewer, thank you for the comment. As we responded in the above comment, the Ultrasound scan was done at the end of first trimester and at second trimester of pregnancy since we included women in these gestational ages. 

Reviewer comment: In page 9,line 140-the authors should explain what baja is.

Authors’ response: Thank you dear reviewer we explained it in the text; Bajaj is a ‘three wheeled vehicle’.

---

## [Decision Letter · Decision Letter 1]

7 Jun 2022

PONE-D-21-12390R1Duration of inter-pregnancy interval and its predictors among pregnant women in urban South Ethiopia: Cox gamma shared frailty modelingPLOS ONE

Dear Dr. Jena,

Thank you for submitting your manuscript to PLOS ONE. After careful consideration, we feel that it has merit but does not fully meet PLOS ONE’s publication criteria as it currently stands. Therefore, we invite you to submit a revised version of the manuscript that addresses the points raised during the review process.

We look forward to receiving your revised manuscript.

Kind regards,

Aniekan Abasiattai

Guest Editor

PLOS ONE

Journal Requirements:

Reviewers' comments:

Reviewer's Responses to Questions

**Comments to the Author**

1. If the authors have adequately addressed your comments raised in a previous round of review and you feel that this manuscript is now acceptable for publication, you may indicate that here to bypass the “Comments to the Author” section, enter your conflict of interest statement in the “Confidential to Editor” section, and submit your "Accept" recommendation.

Reviewer #1: (No Response)

Reviewer #2: (No Response)

2. Is the manuscript technically sound, and do the data support the conclusions?

Reviewer #1: Yes

Reviewer #2: Yes

3. Has the statistical analysis been performed appropriately and rigorously? 

Reviewer #1: Yes

Reviewer #2: I Don't Know

4. Have the authors made all data underlying the findings in their manuscript fully available?

Reviewer #1: Yes

Reviewer #2: Yes

5. Is the manuscript presented in an intelligible fashion and written in standard English?

Reviewer #1: Yes

Reviewer #2: No

6. Review Comments to the Author

Reviewer #1: Some comments also

1. The finding of cluster effect needs to be added in the multivariable Cox gamma shared frailty model result table, in the discussion, abstract and conclusion as it has policy implication.

2. In the predictors section theta value better not more than three digits.

3. Simply base on full model variance of theta and its 95 CI calculate the median hazard with 95%CI and put it at the end of multivariable analysis and interpret it in short, then better to remove the cluster effect analysis section after table 4.

(MHR=1.30 (95CI%,1.11-1.43))

Reviewer #2: GENERAL: There are still several grammatical errors and wrong use of tenses that need to be corrected throughout the text.

ABSTRACT: The last sentence in the results sub-section is rather confusing and should be recasted.

ITRODUCTION: Page 4, lines 1-2,it is not clear what the authors mean by pregnancies that were too long or short. Are they referring to duration of pregnancy or inter-pregnancy interval? In page 4, paragraph 2,line 1, it is not clear which recommendation the authors are referring to and the recommendation should be referenced.

METHODS: Under sample size and data collection- line 4-5, the calculation of inter-pregnancy interval seams to be different from that in lines 1-3 under variables and measurements. When and where were the face to face interviews conducted and how were the identified houses selected?

DISCUSSION: Several grammatical errors. Page 23, paragraph 2, the last sentence should be recasted.

7. PLOS authors have the option to publish the peer review history of their article (what does this mean?). If published, this will include your full peer review and any attached files.

Reviewer #1: No

Reviewer #2: No

---

## [Author Response · Author response to Decision Letter 1]

9 Jun 2022

To: PLOS ONE Journal, Editorial Office

Subject: Submitting a revised version of manuscript and response to Reviewers.

Ref: Submission ID PONE-D-21-12390R1

Title of Article: " Duration of inter-pregnancy interval and its predictors among pregnant women in urban South Ethiopia: Cox gamma shared frailty modeling"

Authors:

Belayneh Hamdela Jena, Gashaw Andargie Biks, Yigzaw Kebede Gete, Kassahun Alemu Gelaye

We would like to thank the Editor for facilitating and giving the opportunity to revise our manuscript. We are also grateful to reviewers for sharing their views and constructive comments. The comments are very important which will improve the quality of our manuscript. The point-by-point responses for each of the comments and the revised manuscript are provided in the attached documents. 

Regards,

The authors!

Editor’s comments:

Journal Requirements:

Authors’ response: Thank you dear Editor for your time and contribution. We checked the references and no retracted references found.

Reviewers’ comments

Reviewer #1

Some comments also

1. The finding of cluster effect needs to be added in the multivariable Cox gamma shared frailty model result table, in the discussion, abstract and conclusion as it has policy implication.

Authors’ response: Thank you dear reviewer for your contribution. We acknowledge your comments. In the revised manuscript we added the cluster effect in all sections according to your suggestion (kindly see the highlighted text in the table, discussion, abstract and conclusion sections of the revised manuscript).

2. In the predictors section theta value better not more than three digits.

Authors’ response: Thank you dear reviewer now we have corrected the digits into three

3. Simply base on full model variance of theta and its 95 CI calculate the median hazard with 95%CI and put it at the end of multivariable analysis and interpret it in short, then better to remove the cluster effect analysis section after table 4.

(MHR=1.30 (95CI%,1.11-1.43))

Authors’ response: Thank you dear reviewer we have calculated the MHR and its 95% CI from full model variance of theta, and put the interpretation in short, under the multivariable analysis. We have deleted the cluster effect analysis section as well.

Reviewer #2: 

GENERAL: There are still several grammatical errors and wrong use of tenses that need to be corrected throughout the text.

Authors’ response: Thank you dear reviewer for your contribution. We acknowledge your comments. In the revised manuscript we have attempted to correct those errors throughout the text as much as possible. 

ABSTRACT: The last sentence in the results sub-section is rather confusing and should be recasted.

Authors’ response: Thank you dear reviewer now we have corrected it. The MHR is normally interpreted in that manner.

ITRODUCTION: Page 4, lines 1-2,it is not clear what the authors mean by pregnancies that were too long or short. Are they referring to duration of pregnancy or inter-pregnancy interval? In page 4, paragraph 2,line 1, it is not clear which recommendation the authors are referring to and the recommendation should be referenced.

Authors’ response: Thank you dear reviewer for suggesting to correct those vague words. In the revised manuscript we have corrected them. Too long and too short were referring to inter-pregnancy intervals. We have also clarified which recommendation we were referring, and cited as well. It refers world health organization recommendation of at least 24 months.

METHODS: Under sample size and data collection- line 4-5, the calculation of inter-pregnancy interval seams to be different from that in lines 1-3 under variables and measurements. When and where were the face to face interviews conducted and how were the identified houses selected?

Authors’ response: Thank you dear reviewer for the comments. We approximated inter-pregnancy interval from birth interval estimate, by subtracting 9 months of average gestational age, which is possible when information for inter-pregnancy interval is lacking, just to estimate sample size. For instance, birth interval 33 months is approximately equivalent to inter-pregnancy interval 24 months (33-9 = 24). Birth interval is could be estimated by subtracting the date of preceding childbirth (delivery) from the date of the recent childbirth (birth interval = date of recent childbirth minus date of preceding childbirth). Inter-pregnancy interval = date of last menstrual period minus date of the recent childbirth. The difference between birth interval and inter-pregnancy interval is the ‘gestational age’. Inter-pregnancy interval estimation does not include gestational age. So inter-pregnancy interval could be estimated from birth interval by subtracting the gestational age when information about inter-pregnancy interval is lacking. This was done because we lacked an inter-pregnancy interval estimates to calculate the sample size during design of the study. However, for our study we have both dates (date of last childbirth and date of last menstrual period or estimate from Ultrasound) information, and the population were pregnant women. The gestational age of women was also known so it is possible to estimate inter-pregnancy interval without birth interval information.

We have done a house-to-house visits in the urban community, and the face-to-face interviews were conducted during the visits, at the house-hold level. For this study, we included all eligible pregnant women who are willing to participate, during the study period. No sampling of houses was done, rather we considered all eligible pregnant women who meet predefined inclusion criteria.

DISCUSSION: Several grammatical errors. Page 23, paragraph 2, the last sentence should be recasted.

Authors’ response: Thank you dear reviewer we attempted to correct the grammatical errors in all sections of the manuscript. We preferred to remove the sentence as it is non-sensing to discuss in that manner.

 END!

---

## [Decision Letter · Decision Letter 2]

12 Jul 2022

Duration of inter-pregnancy interval and its predictors among pregnant women in urban South Ethiopia : Cox gamma shared frailty modeling

PONE-D-21-12390R2

Dear Dr. Jena,

We’re pleased to inform you that your manuscript has been judged scientifically suitable for publication and will be formally accepted for publication once it meets all outstanding technical requirements.

Kind regards,

Aniekan Abasiattai

Guest Editor

PLOS ONE

Additional Editor Comments (optional):

The authors have made the required corrections.
---

## [Editor Report · Acceptance letter]

22 Jul 2022

PONE-D-21-12390R2 

Duration of inter-pregnancy interval and its predictors among pregnant women in urban South Ethiopia: Cox gamma shared frailty modeling 

Dear Dr. Jena:

I'm pleased to inform you that your manuscript has been deemed suitable for publication in PLOS ONE. Congratulations! Your manuscript is now with our production department. 

Kind regards, 

on behalf of

Dr. Aniekan Abasiattai 

Guest Editor

PLOS ONE